# The Connective Tissue Disorder Associated with Recessive Variants in the *SLC39A13* Zinc Transporter Gene (Spondylo-Dysplastic Ehlers–Danlos Syndrome Type 3): Insights from Four Novel Patients and Follow-Up on Two Original Cases

**DOI:** 10.3390/genes11040420

**Published:** 2020-04-14

**Authors:** Camille Kumps, Belinda Campos-Xavier, Yvonne Hilhorst-Hofstee, Carlo Marcelis, Marius Kraenzlin, Nicole Fleischer, Sheila Unger, Andrea Superti-Furga

**Affiliations:** 1Division of Genetic Medicine, Lausanne University Hospital (CHUV), 1011 Lausanne, Switzerland; camille.kumps@chuv.ch (C.K.); belinda.xavier@chuv.ch (B.C.-X.); sheila.unger@chuv.ch (S.U.); 2Department of Clinical Genetics, Leiden University Medical Centre, 2333 ZA Leiden, The Netherlands; y.hilhorst-hofstee@lumc.nl; 3Department of Human Genetics, Radboud University Nijmegen Medical Center, 6525 GA Nijmegen, The Netherlands; carlo.marcelis@radboudumc.nl; 4Clinic for Endocrinology, Diabetes & Metabolism, University Hospital Basel, 4031 Basel, Switzerland; marius.kraenzlin@unibas.ch; 5FDNA Inc., Boston, MA 02111, USA; nicole@fdna.com

**Keywords:** Ehlers–Danlos syndrome, SLC39A13, dysmorphology, short stature, connective tissue, DeepGestalt technology

## Abstract

Recessive loss-of-function variants in *SLC39A13,* a putative zinc transporter gene, were first associated with a connective tissue disorder that is now called “Ehlers–Danlos syndrome, spondylodysplastic form type 3” (SCD-EDS, OMIM 612350) in 2008. Nine individuals have been described. We describe here four additional affected individuals from three consanguineous families and the follow up of two of the original cases. In our series, cardinal findings included thin and finely wrinkled skin of the hands and feet, characteristic facial features with downslanting palpebral fissures, mild hypertelorism, prominent eyes with a paucity of periorbital fat, blueish sclerae, microdontia, or oligodontia, and—in contrast to most types of Ehlers–Danlos syndrome—significant short stature of childhood onset. Mild radiographic changes were observed, among which platyspondyly is a useful diagnostic feature. Two of our patients developed severe keratoconus, and two suffered from cerebrovascular accidents in their twenties, suggesting that there may be a vascular component to this condition. All patients tested had a significantly reduced ratio of the two collagen-derived crosslink derivates, pyridinoline-to-deoxypyridinoline, in urine, suggesting that this simple test is diagnostically useful. Additionally, analysis of the facial features of affected individuals by DeepGestalt technology confirmed their specificity and may be sufficient to suggest the diagnosis directly. Given that the clinical presentation in childhood consists mainly of short stature and characteristic facial features, the differential diagnosis is not necessarily that of a connective tissue disorder and therefore, we propose that *SLC39A13* is included in gene panels designed to address dysmorphism and short stature. This approach may result in more efficient diagnosis.

## 1. Introduction

The Ehlers–Danlos syndrome (EDS) is a group of disorders that affect connective tissues in the skin, ligaments, joints, blood vessels, and other organs. Defects in connective tissue cause a wide range of clinical manifestations, from mildly loose joints to life-threatening conditions, such as arterial bleeds. From the original description as a disorder of hyperelastic skin and lax joints at the beginning of the 20th century [1], a first classification with four main types was proposed in 1970 [2]; molecular advances have allowed the recognition of many distinct disorders that, although different from the classic EDS types described by Beighton, have been given the moniker of “Ehlers–Danlos syndrome” as a reflection of the presence of connective tissue fragility. Thus, the recent version of EDS classification has been expanded to include a wide range of disorders, including skeletal dysplasia [3]. However, the clinical criteria remain relatively non-specific, and clinical diagnosis can be difficult.

A connective tissue disorder associated with recessive biallelic variants in *SLC39A13*, and a mouse knock-out model for the same gene, were described in 2008 by two separate but collaborating groups [4,5]. The features noted in these eight patients were postnatal-onset short stature, protuberant eyes with bluish sclerae and down-slanting palpebral fissures, thin and moderately hyperelastic skin with bruisability, and hands with finely wrinkled palms, tapering fingers, thenar atrophy, and moderate hypermobility of the small joints. Skeletal radiographs showed a moderate degree of platyspondyly with irregular vertebral end plates as well as minor epimetaphyseal changes in appendicular bones. A reduced molar ratio of pyridinoline-to-deoxypyridinoline in urine was observed, indicating reduced collagen lysyl hydroxylation. The latter finding is typically observed in collagen lysyl hydroxylase deficiency (EDS type VI-A), but in that condition, there is severe muscular hypotonia from birth, progressive kyphoscoliosis, and normal height (apart from kyphoscoliosis). Additionally, there were no pathogenic mutations in *PLOD1*; instead, the patients reported by Giunta et al. (2008) were homozygous for an in-frame 9-bp deletion in *SLC39A13*, c.483_491del (p.F162_164del), while those reported by Fukada et al. (2008) were homozygous for the *SLC39A13* variant c.221G>A (p.G74D). The novel condition was given the name of “spondylo-cheiro-dysplastic EDS” for the distinguishing features of the hand and the platyspondyly [4,5]. Later, the EDS nosology has used the name “spondylodysplastic EDS” for a group of three conditions, B3GALT6 deficiency (better known as spondyloepimetaphyseal dysplasia with joint laxity, Beighton type), B4GALT7 deficiency, and SLC39A13 deficiency (the original spondylo-cheiro-dysplastic type) [3]. Of note, these three conditions are also included in the 2019 revision of the skeletal dysplasia nosology [6]. SLC39A13 deficiency is rare and following the initial eight affected individuals, only a single case report has been published [7]. We present a follow up of two of the original patients [5] as well as clinical, radiological, and genetic findings on four new patients from three families.

## 2. Materials and Methods

All affected individuals and/or their legal representatives in the study gave their informed consent to the use of their clinical data, as well as for molecular studies in a diagnostic context (see below).

### 2.1. Molecular Analysis 

Molecular studies for patients 1 and 2 were done in the Lausanne laboratory as described [5]. Molecular studies in patients 3 to 6 were done for diagnostic purposes with appropriate informed consent from patients and their guardians. The studies were done using routine diagnostic sequencing procedures in certified diagnostic laboratories: Analysis for patients 3, 5, and 6 was done in Lausanne, while the analysis of patient 4 was done in Nijmegen. 

### 2.2. Analysis of Pyridinium Crosslink Products by HPLC

Spot urine samples were acid hydrolyzed and analyzed by reverse-phase HPLC as previously described [4].

### 2.3. Analysis of Facial Features

Anonymized frontal facial photographs of individuals with confirmed biallelic *SLC39A13* variants were used to capture the facial gestalt of SLC39A13 deficiency and to compare it to individuals who are clinically normal. All images were fully de-identified through the use of the DeepGestalt image analysis technology [8]. De-identified data of images from 8 patients (the six patients reported here, plus two of the patients described by Giunta et al. (2008) whose photographs had been submitted to us for clinical consultation) were uploaded to the Face2Gene Research app [9] and matched to controls by age, sex, and ethnicity to produce artificial composite images. Because of the small number of patients, the comparison was run twice [10] with two different sets of matched controls. The comparison and separation quality between the three groups was evaluated by measuring the area under the curve (AUC) of the receiver operating characteristic (ROC) curve. To estimate the statistical power of DeepGestalt in distinguishing affected individuals from controls, a cross-validation scheme was used, including a series of binary comparisons between all groups. For these binary comparisons, the data was split randomly multiple times into training sets and test sets. Each such set contained half of the samples from the group, and this random process was repeated 10 times [8,10]. The results of the binary comparisons were reported both numerically and graphically.

### 2.4. Clinical Reports 

#### 2.4.1. Individuals 1 and 2 (reported in part by Fukada et al., 2008 [5]) 

The parents, of Portuguese origin, were of average stature and clinically unremarkable. They were not knowingly related, although molecular workup showed that they shared a common haplotype harboring the *SLC39A13* pathogenic variant [5]. The affected children, a boy and a girl, were born at term from uncomplicated pregnancies and were of normal size and weight at birth but showed progressive short stature beginning in the second half of the first year of life. Among the clinical signs in early childhood were muscular hypotonia and soft skin, leading to the diagnostic suspicion of the Ehlers–Danlos syndrome (EDS). During childhood, the main clinical signs and features were thin, fragile, but not hyperelastic skin that bruised easily and was particularly thin on the hands and feet; varicose veins; moderate joint laxity; blueish or greyish sclerae; down-slanting palpebral fissures; and the absence of one or more teeth in permanent dentition. Both had astigmatism in childhood. Radiographic examination revealed moderate osteopenia, flattened or biconcave vertebral bodies with flaky irregularity of the endplates as well as mild dysplastic changes at the metaphyses of long bones and of the phalanges. At the time of the first report, they were 28 years old and 145 cm tall (individual 1, male); and 20 years old and 135 cm tall (individual 2, female). Their body proportions were normal, indicating that the platyspondyly was accompanied by shortening of the long bones. Their skin remained thin and fragile, and the subcutaneous fat tissue was sparse. Both individuals had marked venous varicosities on their legs and feet. Individual 1 suffered from a cerebral hemorrhage posteriorly to the left putamen at age 22 years, from which he recovered completely. He has successfully completed higher education.

Subsequent to the initial publication in 2008, individual 1 has had no further complications; he is professionally active, has married, and his wife has given birth to a healthy child. Individual 2, his younger sister, had obtained higher education degrees at age 22 and 25 years. At age 25 years, cerebral vascular imaging was obtained, and no abnormalities were observed. At age 26 years, two weeks after the interruption of hormonal contraception prescribed because of irregular cycles, she suffered from arterial thromboembolism that caused cerebral ischemia with right arm paresis and Broca’s aphasia. Several days after initiation of aspirin therapy, she developed cerebral hemorrhage leading to complete right hemiparesis. Fortunately, she was able to partially recover over four years on physiotherapy but still has partial function of the right hand and myoclonus on the right arm. She received speech/language therapy; however, she still has some degree of dysphasia. Nevertheless, she was able to return to her previous work.

#### 2.4.2. Individual 3

This is a girl born by caesarean section at 33 weeks of gestation from a dizygotic twin pregnancy. Both the patient and her twin sister were 2500 g at birth. The patient developed progressive short stature over her first years of life, while her twin sister grew normally. Motor and intellectual development were normal. The parents are first cousins of Turkish origin and measure 148 cm and 170 cm with normal proportions. An elder brother had normal growth and body proportions. During her teenage years, she developed severe keratoconus with no complications so far. She complains about back pain and her skin bruises easily, with hematomas and blood blisters at sites of friction. Radiographic examination of the spine at age 15 revealed platyspondyly of the thoracic and lumbar vertebrae with mild anterior beaking. On physical examination at age 23, she had short stature. There was moderate diffuse joint laxity most marked on the finger joints. The skin on her hands is thin and wrinkled and she has tapering fingers with narrow end phalanges. Facial features include downslanting palpebral fissures, a flat face, protuberant eyes with reduced periocular tissue, greyish sclerae, and a small mouth. 

#### 2.4.3. Individual 4

This boy was the fourth child from healthy parents of Turkish descent. The parents were second cousins and measured 165 cm (father) and 158 cm (mother). His three older sibs have normal growth. He was born after an uneventful term pregnancy with a birthweight of 3500g. He developed progressive short stature over his first years of life and was first seen by a pediatrician at age 8 1/2 years. Height was 113.4cm (−3.8 SDS) and weight was 23.5kg (−1.5 SDS). Psychomotor development was normal. He complained of knee and ankle pain after exercise. He had mild weakness of the hands, making it difficult to open bottles. He had no clear dysmorphic features, but his palpebral fissures were down-slanting and he had oligodontia (missing five teeth). He has had no eye complication so far. He had a short trunk and stands with lumbar hyperlordosis. He had loose skin on hands and feet, moderate hyperlaxity of fingers and toes, valgus deformity of the ankles, and bilateral pes planus. Radiological examination showed platyspondyly of the thoracic and lumbar spine with mild anterior beaking and mild metaphyseal irregularities especially of the distal ulna. He first received a clinical diagnosis of spondylometaphyseal dysplasia. Molecular analysis of *SHOX*, *FGFR3*, *TRPV4*, and *LTBP3* were negative, but further studies showed a biallelic missense change in *SLC39A13* (Table 1). Additional urine analysis showed a low ratio of pyridinoline-to-deoxypyridinoline (Figure 3).

#### 2.4.4. Individuals 5 and 6 

The family originates from Syria and access to early medical records is impossible. The parents, who are first cousins, are clinically healthy and measure 150 cm in height (mother) and 160 cm (father). The proband (patient 6; Figure 1) is a 9-year-old boy referred for evaluation for small stature. We found that his elder sister is similarly affected (patient 5), while a younger sister is clinically normal. Individual 6 was born at term after an uneventful pregnancy, although IUGR was noted and the mother reports reduced fetal movements throughout the pregnancy. Birth data were not recorded, but the mother remembers his length to be reduced. At birth, the mother noted the he had no spontaneous movements, had edema on both feet, and increased palmar creases on his hands and feet. Throughout childhood, psychomotor development was slightly delayed, but intelligence and behavior at age 9 years were normal. At this age, his growth parameters were height 111 cm (−3.9 SD), weight 19 kg (−3.3 SD), and head circumference 51.5 cm (−0.78 SD). He had down-slanting palpebral fissures, protruding eyes, hypertelorism, and a large forehead. An ophthalmologic evaluation revealed bilateral myopia and marked keratoconus. Hearing was normal. Four teeth of the permanent dentition were found to be missing. His elbow joints appeared prominent, but he had no other signs of joint deformity or scoliosis. He had hyperlaxity of small joints and complained occasionally of joint pain particularly in his knees. The skin was thin and wrinkled, particularly on the palmar side of his hands and feet, and the thenar muscles were underdeveloped. Radiographic studies showed small epiphyses and posterior subluxation of the radii and mild platyspondyly. Routine blood tests were unremarkable. Analysis of *SHOX* done prior to referral was negative.

Individual 5 is the older sister of individual 6. According to the birth date declared at immigration, she would be 10 years old. However, we doubt the accuracy of this date and suspect that she is older than the stated age. Birth parameters have not been recorded. Her growth parameters were as follows: height 131.5 cm (−1 SD for age 10), weight 29 kg (−0.81 SD for age 10), and head circumference 51.5 cm (−0.55 SD for age 10). Shortly after she was referred, she had menarche and her growth has slowed down significantly. She had protruding eyes and a large forehead, but her facial appearance was less conspicuous than that of her brother or other affected individuals. She had myopia, and four incisor teeth were missing. Intelligence and hearing were normal. She had long tapering fingers and her interphalangeal joints were hyperextensible. The skin was thin and wrinkled, particularly on the palmar side of her hands and feet. 

## 3. Results

Figure 1 shows some clinically relevant clinical and radiographic features in SLC39A13-deficient individuals. 

Table 1 shows the genetic variants identified in *SLC39A13*. Of note, the variants were homozygous in all patients, underlining the rarity of the disorder. 

Figure 2 shows the results of the automated DeepGestalt technology. Both comparisons (*SLC39A13* group vs. control group 1, and *SLC39A13* group vs. control group 2) gave highly significant results (AUC= 0.991 p = 0.006 and AUC= 0.986 *p* = 0.004).

Figure 3 shows the ratio of the collagen crosslink derivates, pyridinoline and deoxypyridinoline, in the urine of five patients and five roughly age-matched controls. The ratio of pyridinoline-to-deoxypyridinoline is reduced in the SLC39A13-deficient subjects, indicating a relative deficiency of the hydroxylated form, pyridinoline, relative to the non-hydroxylated form, deoxypyridinoline. The values clearly segregated in two distinct clusters with no overlap. In addition to confirming the notion of reduced lysyl hydroxylation as a pathogenic mechanism in these patients, the findings show that determination of urinary crosslink derivates can be a useful diagnostic screening method. Anecdotally, because of the role of vitamin C in collagen hydroxylation, individuals 1 and 2 took an oral dose of 1000 mg of vitamin C over four weeks and sent us urine samples for analysis before and twice during the test period. Unfortunately, there was no change in the pyridinoline-to-deoxypyridinoline ratio. 

## 4. Discussion

### 4.1. Clinical Aspects of SCL39A13 Deficiency

Our observations confirm the previous findings of Giunta et al. (2008), Fukada et al. (2008), and Dusanic et al. (2018) that SLC39A13 deficiency is associated with a clinical phenotype characterized by post-natal short stature, connective tissue weakness affecting mainly the skin and peripheral joints, a characteristic facial appearance, and a moderate skeletal dysplasia. These additional observations help to delineate a more precise phenotype. 

**Muscular hypotonia**—This has been seen in several *SLC39A13* patients in the neonatal period and in early childhood; in the patient described by Dusanic et al. (2018), muscular hypotonia was conspicuous enough to lead to investigations for myopathy with some abnormal (though non-specific) findings in a muscle biopsy at adolescence. While the prevalence of myopathy in SLC39A3 deficiency and its structural features remain to be investigated, the observations confirm the well-known phenomenon that connective tissue diseases, and particularly EDS type VI-A (lysyl hydroxylase deficiency), may present as the “floppy infant” phenomenon and/or with hypotonia with delayed motor development. 

**Growth and stature**—postnatal reduced linear growth seems to be a salient and consistent feature of SLC39A13 deficiency; this is in contrast to other types of connective tissue disease and EDS types, where short stature is not a prominent feature. The two other forms of “spondylodysplastic EDS”, namely B3GALT6 deficiency and B4GALT7 deficiency, are also associated with short stature, but in those conditions, short stature can be explained by skeletal dysplasia. In *SLC39A13*, body proportions are normal and short stature seems to be more intrinsic and represents a true growth failure and not a consequence of bone dysplasia.

**Eye findings**—Myopia has been reported in several *SLC39A13* patients and also seems to be a direct (though non-specific) manifestation of the primary genetic defect. This may be connected to the thin sclerae that often have a blueish or greyish color, particularly in individuals with darker pigmentation. Keratoconus, as seen in three patients, is potentially dangerous because of the possibility of perforation or rupture; ophthalmologic investigation should be recommended in all confirmed cases. 

**Dental features**—Absence of one or several incisor teeth in the permanent dentition has been observed in a majority of SLC39A13-deficient individuals. In our case 6, the lower incisors of the primary dentition were also small and dysplastic. Hypodontia and oligodontia are consistent, if perhaps not obligate, features of the disorder. 

**Vascular complications**—marked varicosities of the lower legs have been described in a number of *SLC39A13* patients. Clinically, more importantly, the two elder patients known (patients 1 and 2 in Fukada et al. (2008) and in this study) have both suffered cerebral hemorrhage. While this may not be statistically significant and may even be causally unrelated (e.g., hormonal contraception in patient 2), it is worth keeping in mind as a potential complication. Fortunately, other major vascular events have thus far not been recorded. 

**Facial features**—analysis of the facial features with the DeepGestalt technology and comparison with age-matched controls confirms that there is an SLC39A13-associated facial phenotype that is significantly divergent from that of control individuals. This phenotype includes a rather flat face, mild hypertelorism, downslanting palpebral fissures, lack of periocular connective tissue giving the impression of prominent eyes, and a small mouth. The Face2Gene system evoked a few other syndromes with some resemblance to the *SLC39A13* phenotype, such as Noonan syndrome (downslanting palpebral fissures), Stickler syndrome (flat face; associated with *COL2A1* haploinsufficiency), as well as—interestingly—the vascular type of EDS (EDS IV; dominant *COL3A1*), probably because of the “hollow eyes”. Thus, analogy between SLC39A13 deficiency and vascular EDS may include vascular fragility as well as a moderate resemblance of facial features. 

### 4.2. What Are the Pathogenetic Mechanisms Leading from SLC39A13 Loss of Function to the Complex Clinical Phenotype?

The pathogenesis of SLC39A13 deficiency remains poorly understood. The observation of reduced collagen hydroxylation indicates that a partial failure of collagen crosslinking is one pathogenic mechanism [4], and the determination of pyridinoline-to-deoxypyridinoline is a useful diagnostic help. As correctly indicated by Giunta et al. (2008), the hydroxylation defect probably involves collagens other than collagen 1, such as collagen 2 and collagen 3, and this may explain the mild chondrodysplastic features (collagen 2) as well as the thin and fragile skin and the putative vascular fragility (collagen 3). However, it is difficult to ascribe the short stature, facial features, and oligodontia solely to reduced collagen hydroxylation. The hypothesis that part of the pathogenesis may involve the role of zinc as an intracellular messenger [5] may be justified, but experimental evidence is lacking; in particular, SLC39A13 individuals do not show signs of immune system dysfunction in contrast with the mouse model [5,11].

### 4.3. Diagnosis

From the diagnostic perspective, presentation with short stature and dysmorphic facial features may lead pediatricians and clinical geneticists to classify these patients within the group of genetic syndromes rather than connective tissue disorders; this “syndromic” presentation of SLC39A13 is not well known and this report may help raise awareness of this condition and its presenting features among clinical geneticists and dysmorphologists. 

## Figures and Tables

**Figure 1 genes-11-00420-f001:**
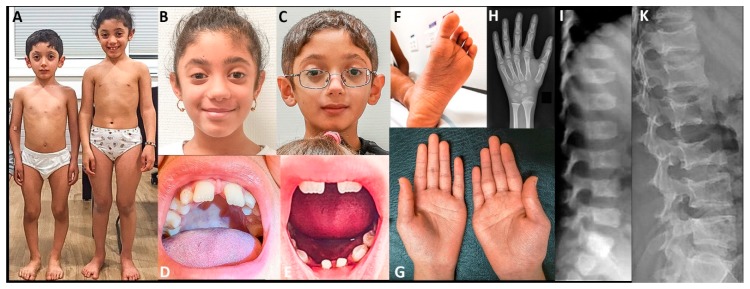
Clinical and radiographic findings. Individuals 5 (panels **A** and **B**) and 6 (panels **A** and **C**) at the age of eight and ten years. Panels **D** and **E**: missing upper incisives, and missing lower incisives with persistence of primary teeth in individuals 6 at age 9 years. Panel **F**: foot of individual 6 showing finely wrinkled skin and abnormally deep furrows. Panel **G**: Palmar aspects of the hands of individual 5 showing thin skin with increased number of fine wrinkles and mild thenar and hypothenar atrophy. Panel **H** shows the hand radiograph of individual 5 at age ten years. There is mild diaphyseal overconstriction of radius and ulna, metacarpals, and phalanges; bone maturation is roughly appropriate; overall, the changes are mild and non-diagnostic. Panels **I** and **K**: lateral spine radiographs of individual 3. At age 6 years, there is moderate platyspondyly. At age 15, the end plates have a marked concave conformation.

**Figure 2 genes-11-00420-f002:**
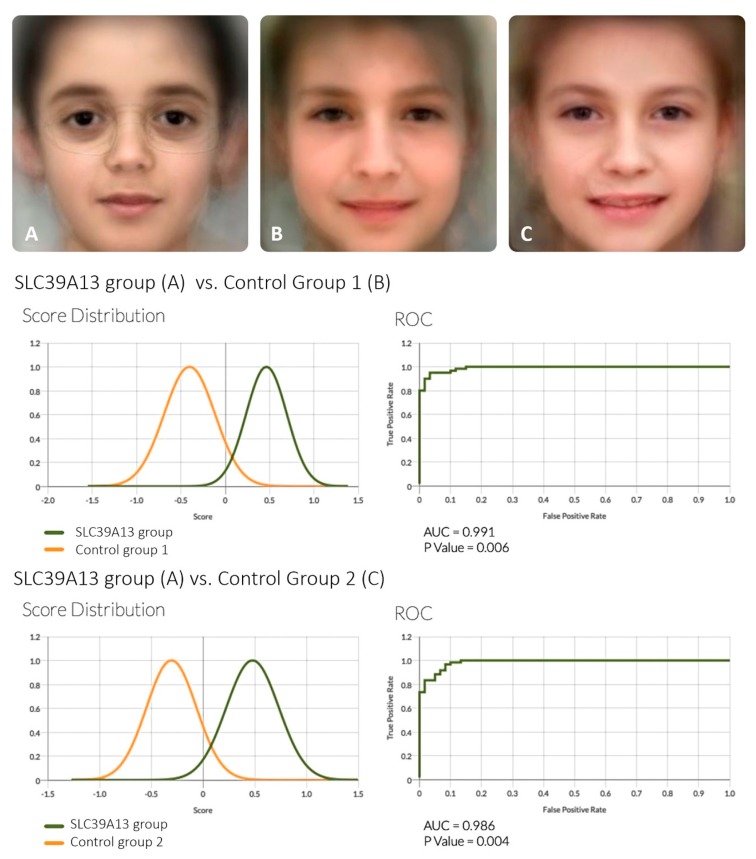
DeepGestalt technology analysis of facial features in SLC39A13 deficiency. The upper part shows the “averaged” artificial composite facial gestalt image of individuals with SLC39A13 deficiency (panel **A**) vs. two matched control groups (panels **B** and **C**). The lower part shows the result of the comparisons between **A** vs. **B**, and **A** vs. **C**.

**Figure 3 genes-11-00420-f003:**
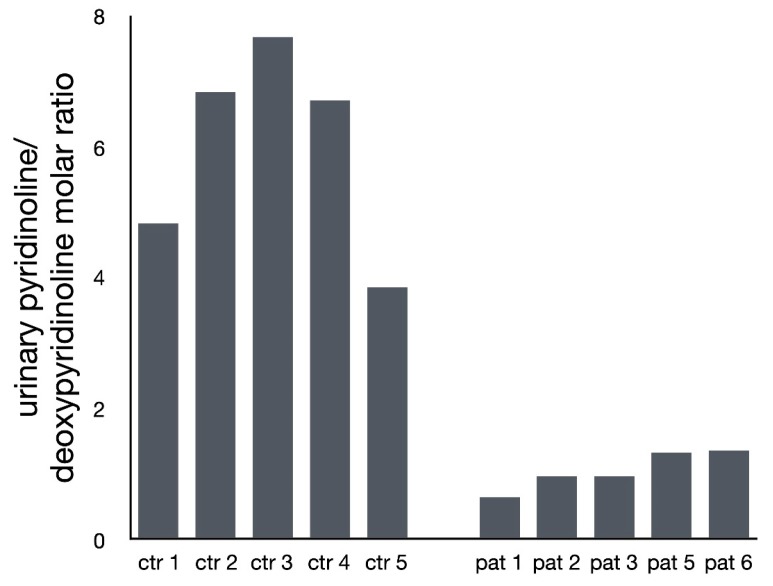
Ratio of the collagen crosslink derivates, pyridinoline and deoxypyridinoline, in the urine of five SLC39A3-dificient individuals and in five roughly age-matched controls. The ratio is consistently lower in SLC39A13-deficient individuals, with no overlap to the control group.

**Table 1 genes-11-00420-t001:** Overview of the molecular results.

	Pat. 1 *	Pat. 2 *	Pat. 3	Pat. 4	Pat. 5	Pat. 6
***SLC39A13* (NM_001128225.3)**	c.G221A/c.G221A	c.G221A/c.G221A	c.483_491delCTTCCTGGC/c.483_491delCTTCCTGGC	c.793G>A/c.793G>A	c.1019delT/c.1019delT	c.1019delT/c.1019delT
**Protein (NP_001121697.2)**	p.Gly74Asp	p.Gly74Asp	p.Phe162Ala164del	p.Asp265Asn	p.Leu340ProfsTer23	p.Leu340ProfsTer23
**Frequency in gnomAD ****	absent	absent	extremely low, AF = 0.000009	absent	absent	absent
**Predictors of pathogenicity on protein *****	pathogenic	pathogenic	pathogenic	pathogenic	pathogenic	pathogenic

Notes: * these two siblings have been reported by us before (Fukada et al., 2008); ** as assessed in March 2020 on the gnomAD database (https://gnomad.broadinstitute.org/); *** as assessed in March 2020 on the VARSOME integrative website (https://varsome.com/).

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
