# Peer review of "The Connective Tissue Disorder Associated with Recessive Variants in the SLC39A13 Zinc Transporter Gene (Spondylo-Dysplastic Ehlers–Danlos Syndrome Type 3): Insights from Four Novel Patients and Follow-Up on Two Original Cases"

_genes, 2020, doi:10.3390/genes11040420_

Round 1
Reviewer 1 Report
This is a very interesting and well written paper on the ultrarare SLC39A13-related spondylodysplastic type of Ehlers-Danlos syndrome. Dr. Krumps and colleagues report four previously unpublished cases and the follow-up of two of the originally reported patients, from a total of nine published individuals. The clinical description is complete and interesting thanks to the heterogeneity of the age at referral. In addition, three of the identified variants are novel.
The reviewer’s main comment concerns nosology of this condition. The authors reasonably state that the combination of short stature and facial dysmorphism is quite unusual for Ehlers-Danlos syndrome(s); hence, they suggest to classify this disorder within the group of “genetic syndromes” rather than among the “(hereditary) connective tissue disorders”. Classifying SLC39A13-related phenotypes is not an easy task. To support this assertion there are (i) the very limited number of reported patients to date and (ii) the presence of this condition in at least two parallel nosologies: the 2017 classification of Ehlers-Danlos syndrome and related disorders (Malfait et al., 2017), and the nosology and classification of genetic skeletal disorders (Mortier et al., 2019). The existence of a significant clinical overlap between disorders with prevalent involvement of the non-ossified connective tissues and those with predominant skeletal manifestations is testified by the inclusion of various rare and ultrarare variants of Ehlers-Danlos syndromes into the nosology and classification of genetic skeletal disorders (CHST14, DSE, SLC39A13, PLOD1, FKBP14, B4GALT7, B3GALT6). This phenomenon is relevant. In fact, a wider spectrum of disorders classically considered predominantly affecting the non-ossified connective tissues, such as Marfan syndrome and the other TFGbeta-pathies, have been also included in this nosology. On this perspective, the nosology and classification of genetic skeletal disorders appear as a “meta-nosology” incorporating an extremely variable range of disorders sharing a significant bone involvement but not necessarily dominated by the same pattern(s) of clinically relevant manifestations. Under this perspective, the fact that a disorder may appear in two or more distinct nosologies is acceptable and, perhaps, encouraged.
More strictly concerning SLC39A13-related spondylodysplastic type of Ehlers-Danlos syndrome, accumulated evidence and the eminent opinion of the authors of this paper testifies for a significantly generalized bone involvement (i.e. short stature) in this condition, and this suffices to recognize SLC39A13-related spondylodysplastic type of Ehlers-Danlos syndrome among the genetic skeletal disorders (according to Mortier et al., 2019). On the other hand, the striking high rate of peripheral joint hypermobility and related orthopedic/occupational manifestations, skin texture anomalies, corneal/ocular involvement, muscle hypotonia not clearly secondary to a primary muscle disorder (quite common in many ultrare Ehlers-Danlos sydrome variants), and vascular complications easily put the SLC39A13-related phenotype also under the Ehlers-Danlos syndrome umbrella.
The study that this group carried out on the facial phenotype should be recognized and acknowledged. Although still unpublished, also other and more common Ehlers-Danlos syndrome variants seem to have a subtle facial phenotype. Therefore, it is not excluded that, in the future, similar studies will be carried out also for these variants. In the meanwhile, emphasizing the facial phenotype of SLC39A13-related phenotype is surely useful for rising the attention on this potentially neglected condition. Nevertheless, proposing to list this condition in a third nosology could be too premature, or, at least, could raise more confusion around such a still incompletely defined condition.
Author Response
Dear Reviewer,
Thank you for your input and suggestion.
We fully agree and have inserted a sentence.
Kind regards,
Camille Kumps
Reviewer 2 Report
The authors report on 4 novel patients with an exceedingly rare disorder and provide follow up information on 2 originally reported cases. The authors are experts in the field and are well suited to assess this connective tissue and skeletal disorder. The authors stress that the presentation resembles more a syndrome (with short stature; oligodontia; distinctive facial features) than a connective tissue disorder. In this context, the distinctive facial features, identifiable through AI tools, could provide a useful diagnostic suggestion. The characteristic urinary collagen crosslink derivates could further support such a diagnostic consideration. Thus, the diagnosis could be identified in resource poor countries where exome analysis is not typically available.
Specific recommendations:
- title: change to "....: Insights from four novel patients and follow up on two original cases"
- abstract: omit the term "so far". This is implied. Remove this term throughout the manuscript.
- abstract: edit the last sentence, the term "gene panels" does not need to be repeated. Omit the last semicolon and make a new sentence: "This approach may result in more efficient diagnosis."
- page 2, line 47: "some of which" is poor grammar. Would change this to "...disorders including some skeletal dysplasias"
- page 2 line 50: Omit the term "gene" after the gene name in italics. This is implied. Should be removed throughout the manuscript.
- page 2 line 70: remove the term "subsequent", this is implied in "following the initial eight affected individuals".
- page 2 line 79:change "within a diagnostic itinerary" to "for diagnostic purposes" and omit "for diagnostic testing" at the end of the sentence.
- page 3 line 92: provide a reference or website for the Face2Gene Research app.
- page 3 line 102: Fig. 2 is the first figure mentioned in the text. This should be then be Fig. 1.
- page 4 line 177: providing SD for growth parameters is helpful. The authors should decide whether they use SD or centiles. Indicating "P3-10" is unclear, at least this should be explained as "centiles", ideally with a reference as to which curves were used (WHO?). Preferably, all measurements throughout the manuscript should use the same reference, either SD or centiles.
- page 5 line 187: The sentence "would put her as being 10 years old, however ..." needs editing. Suggest: "Based on her date of birth declared at immigration, she would be 10 years old. However, we doubt the accuracy of this date and suspect that she is older than the stated age."
- Legend Fig. 1: use the term "radiograph", not "Xray".
- page 9 line 287: chnage the term "antimongoloid"- this is inappropriate!
Author contributions; Funding: This manuscript appears poorly prepared because these parts are clearly not filled out or reviewed.
Author Response
Specific recommendations:
- title: change to "....: Insights from four novel patients and follow up on two original cases"
thank you for your suggestion, we changed the title accordingly
- abstract: omit the term "so far". This is implied. Remove this term throughout the manuscript.
Done throughout the manuscript (page 1 lines 20 and 28)
- abstract: edit the last sentence, the term "gene panels" does not need to be repeated.
Done, our apologies (page 1, line 35)
- Omit the last semicolon and make a new sentence: "This approach may result in more efficient diagnosis."
The sentence was corrected (page 1 line 35)
- page 2, line 47: "some of which" is poor grammar. Would change this to "...disorders including some skeletal dysplasias"
The sentence was corrected (page 2, line 63)
- page 2 line 50: Omit the term "gene" after the gene name in italics. This is implied. Should be removed throughout the manuscript.
Done throughout the manuscript (page 1 line 17, page 2 lines 65 and 76, page 4 line 212, page 5 table 1)
- page 2 line 70: remove the term "subsequent", this is implied in "following the initial eight affected individuals".
The sentence was corrected (page 2 line 86)
- page 2 line 79:change "within a diagnostic itinerary" to "for diagnostic purposes" and omit "for diagnostic testing" at the end of the sentence.
The sentence was corrected (page 2 line 94)
- page 3 line 92: provide a reference or website for the Face2Gene Research app.
We provided the website in the references (page 3 line 118)
- page 3 line 102: Fig. 2 is the first figure mentioned in the text. This should be then be Fig. 1.
We erased the mention of Fig 2, see page 3 line 128
- page 4 line 177: providing SD for growth parameters is helpful. The authors should decide whether they use SD or centiles. Indicating "P3-10" is unclear, at least this should be explained as "centiles", ideally with a reference as to which curves were used (WHO?). Preferably, all measurements throughout the manuscript should use the same reference, either SD or centiles.
We switched all measurements to SD, see page 4, lines 204, 216 and 217
- page 5 line 187: The sentence "would put her as being 10 years old, however ..." needs editing. Suggest: "Based on her date of birth declared at immigration, she would be 10 years old. However, we doubt the accuracy of this date and suspect that she is older than the stated age."
We corrected the sentence as suggested, page 4 line 214
- Legend Fig. 1: use the term "radiograph", not "Xray".
We changed the term to radiograph, see page 6 line 283
- page 9 line 287: chnage the term "antimongoloid"- this is inappropriate!
Absolutely correct, please forgive our mistake, we switched it to “downslanting palpebral fissures” see page 9 line 369.
Author contributions; Funding: This manuscript appears poorly prepared because these parts are clearly not filled out or reviewed.
We were asked to fill those out after submission, they are now detailed page 10 lines 400 to 406.